# Adjunctive Procedures in Immediate Implant Placement: Necessity or Option? A Systematic Review and Meta-Analysis

**DOI:** 10.3390/ma18235427

**Published:** 2025-12-02

**Authors:** Isabella De Rubertis, Adriano Fratini, Maria Clotilde Carra, Marco Annunziata, Nicola Discepoli

**Affiliations:** 1Unit of Periodontics, Department of Medical Biotechnologies, Università degli Studi di Siena, 53100 Siena, Italy; isabella.derubertis@gmail.com (I.D.R.); fratini.adriano@gmail.com (A.F.); 2U.F.R. of Odontology, Université Paris Cité, 75006 Paris, France; mclotildecarra@gmail.com; 3Department of Translational Medicine, University of Ferrara, 44121 Ferrara, Italy; 4Multidisciplinary Department of Medical-Surgical and Dental Specialties, University of Campania “Luigi Vanvitelli”, 81100 Naples, Italy; marco.annunziata@unicampania.it

**Keywords:** dental implants, bone substitutes, tooth extraction, tooth socket, grafting, esthetics

## Abstract

**Methods**: Currently available randomized controlled clinical trials (RCTs) reporting on the adjunctive clinical effects of biomaterials, grafting materials, and grafting techniques on immediate implant placement (IIP) were systematically assessed. Data were qualitatively analyzed and, when appropriate, meta-analysis was performed. Risk of bias and quality of evidence were evaluated using the Cochrane RoB 2 tool and the GRADE framework, respectively. **Results**: A total of 12 RCTs (484 implants, 6–36 months of follow-up) were included and consistently reported high implant survival rates (96–100%). Data on the use of xenografts, alloplastic and mixed grafts, connective tissue grafts and socket shield technique seem, to different extents, to demonstrate favorable results in terms of peri-implant hard and soft tissue dynamics and esthetic outcomes. Quantitative synthesis conducted on four RCTs demonstrated significantly higher short-term patient-reported postoperative pain, assessed on a 0–100 Visual Analog Scale (VAS) (weighted mean difference 19.45 mm; 95% CI 0.55–38.36; *p* = 0.04). Most RCTs were rated at moderate to high risk of bias, and certainty of evidence was low to moderate. **Conclusions**: Regardless of the use of adjunctive materials/approaches, IIP guarantees high implant survival rates. Although different adjunctive strategies to IIP may favor hard and soft tissue stability, they appear to increase short-term patient-reported morbidity. Currently available evidence lacks standardized and patient-centered outcome reporting.

## 1. Introduction

Immediate implant placement (IIP) has been shown to be a predictable treatment option for replacing non-restorable teeth [1,2]. Nevertheless, this approach is technically challenging, as it is associated with a higher incidence of biological and esthetic complications [3,4]. From a biological standpoint, the placement of an immediate implant does not inherently limit the alveolar remodeling that occurs following tooth extraction [5,6]. The continuous remodeling process, which reaches its peak during the first 3 months post-extraction, influences the future anatomy of both hard and soft tissues [7]. This biological determinant will condition the future esthetic and functional impact of the site. Indeed, midfacial gingival recession has frequently been reported [3] and resorptive changes in the facial contour are to be expected [8,9].

Dimensional alterations of the facial bone plate are clinically significant, as they encompass both vertical and, more prominently, horizontal resorption [5,10,11]. These changes compromise the supporting function of the buccal bone, directly affecting the overlying soft tissues and significantly impacting both the marginal mucosal level [12] and labial contour. While some studies suggest that early healing may partially compensate for these dimensional changes [13], long-term follow-up studies have demonstrated that reduction in the soft tissue contour is crucial in influencing the esthetic outcome of prosthetic reconstruction. Furthermore, recent randomized controlled trials have confirmed that immediate implant placement alone does not fully prevent post-extraction dimensional changes, with both radiographic and soft tissue analyses revealing significant ridge resorption compared to spontaneous healing [14,15].

To counteract bone resorption, preserve ridge morphology and improve clinical and functional results, several strategies have been proposed in conjunction with IIP. One of the most widely investigated approaches is the use of bone substitute materials (BSMs), such as autografts, allografts, xenografts, or alloplasts. Such biomaterials, positioned in the implant–socket gap, aim to limit bone remodeling, sustain hard tissue three-dimensional stability and improve marginal bone-to-implant contact [5,16,17,18]. Deproteinized bovine bone mineral (DBBM) has shown the ability to limit volumetric changes in the facial alveolus and favor greater tissue stability [8,16].

Additional techniques have been explored to enhance peri-implant tissue preservation, including the socket-shield technique (SST) introduced by Hürzeler et al. (2010) and immediate provisionalization and subepithelial connective tissue grafts (CTGs) [19]. Randomized controlled trials have demonstrated that immediate provisionalization can limit midfacial soft tissue collapse compared with delayed provisionalization [20,21].

However, the literature remains inconclusive. Randomized controlled trials are often limited by small sample sizes, heterogeneous protocols, and inconsistent outcome reporting, frequently leading to conflicting results [22,23,24]. Previous systematic reviews have also concluded that current evidence is insufficient to determine whether grafting the facial gap or applying adjunctive regenerative strategies consistently provide additional benefits in preventing alveolar ridge reduction and contour collapse. Beyond implant survival and hard tissue stability, increasing attention has been directed toward patient-reported and clinically assessed esthetic outcomes. However, despite their clinical relevance, esthetic results are still inconsistently reported, often inadequately and without standardized measures [25,26]. This gap in the literature limits the evaluation of the true benefits of adjunctive procedures.

Therefore, the aim of the present systematic review is to critically assess the clinical effectiveness of biomaterials associated with IIP in the esthetic area in terms of (a) implant survival, (b) esthetic outcomes, and (c) peri-implant soft and hard tissue dimensions.

## 2. Materials and Methods

### 2.1. Protocol Development and Registration

The present systematic review was conducted and reported in accordance with the methodological guidelines of the Cochrane Handbook [27] and the Preferred Reporting Items for Systematic Reviews and Meta-Analyses (PRISMA) 2020 checklist [28,29]. The review protocol was registered a priori on PROSPERO (CRD42025649944).

### 2.2. Criteria Definition for PICOS Framework

The following PICOS framework, incorporating the components of the review research question, was developed [30]:Population: Adults requiring a single dental implant in a post-extraction socket in the esthetic area (maxillary or mandibular premolar-to-premolar region);Intervention: Use of any biomaterial, grafting material, and grafting technique as an adjunctive approach during IIP;Comparison: IIP without the use of any adjunctive material/approach;OutcomesPrimary: Implant survival rate; clinically assessed and patient-reported esthetic outcomes;Secondary: Soft and hard tissue dimensional changes, keratinized tissue changes and self-reported post-operative pain.

Radiographic (periapical or CBCT), clinical and digital/3D analyses for hard and soft tissue parameters were considered appropriate methods for outcome assessment. Validated and standardized indices (Pink esthetic score [PES], White esthetic score [WES]) were accepted for esthetic outcomes evaluation. Post-operative pain scores were included when measured through validated patient-reported scales. Measurements registered on the day of immediate implant placement, before the implant insertion and the application of any adjunctive material, were considered baseline values.

Study Design: Randomized Controlled Trials (RCTs).

### 2.3. Literature Search Restrictions

Eligibility criteria

The following eligibility criteria for study inclusion were established and structured in accordance with the PICO framework:Population: Human participants aged ≥18 years; Patients receiving single immediate dental implants within the esthetic area.Comparison: Immediate implant placement performed without the use of any grafting nor adjunctive material.Outcomes: Studies reporting on at least one of the review’s primary or secondary outcomes.Study design: RCTs; studies published in English; minimum of 10 participants per study group; Minimum follow-up of 6 months after IIP.

### 2.4. Information Sources, Search Strategies and Study Selection

The electronic literature search was conducted independently by two reviewers (AF, IDR) on MEDLINE (PubMed), EMBASE, and Scopus. OpenGray and Google Scholar were also consulted for the gray literature search. References published up to April 2025 were screened. Bibliographies of published reviews and clinical or pre-clinical studies were manually searched and consulted.

Search strategies, adapted to the different databases, are reported in Appendix A. The literature screening process was conducted in a stepwise manner using Nested Knowledge software (https://about.nested-knowledge.com, accessed on 5 February 2025).

Disagreements between reviewers were resolved through discussion and, when consensus could not be reached, by consulting a third author (MCC or ND). Inter-reviewer agreement was assessed by calculating the Cohen’s kappa coefficient (k).

### 2.5. Data Collection Process and Data Items

Data from included articles were extracted by two reviewers (AF, IDR) independently, using a standardized Excel spreadsheet. A third reviewer (ND) verified the appropriateness of the data and developed the final dataset, which was used to generate summary tables for the qualitative synthesis and to compute, when applicable, the quantitative analysis.

Two summary tables were produced. The first table summarizes study characteristics, including year of publication, study setting, participants’ demographic characteristics and clinical features, site-specific and implant characteristics, and the outcomes assessed. The second table reports the numerical results of each study outcome and guided the qualitative synthesis of the present review. Whenever data were unavailable but considered retrievable, the reviewers contacted the corresponding authors; whenever data could not be retrieved, the indication “NR” (not reported) was used. All extracted data items were predefined, including implant survival, marginal bone level changes, horizontal and vertical soft and hard tissue measurements, esthetic outcomes (PES/WES), keratinized tissue width, and postoperative morbidity assessed through validated patient-reported scales.

### 2.6. Risk of Bias Assessment and Certainty Assessment

Version 2 of the Cochrane Risk of Bias tool for randomized controlled trials (RoB 2) [27,31] was utilized to conduct risk of bias assessment. Two reviewers (IDR, AF) performed Rob in duplicate and, as for the study selection process, any disagreement was solved through discussion or a third reviewer (MCC) was consulted to reach consensus.

Moreover, the certainty of the evidence for each outcome was evaluated using the GRADE approach (Grading of Recommendations Assessment, Development and Evaluation) [32,33]. Judgements were made for each of the following domains: risk of bias, inconsistency, indirectness, imprecision, and publication bias.

### 2.7. Data Analysis

Statistical analyses were performed using dedicated statistical software (STATA IC, version 18, StataCorp LP, College Station, TX, USA). A meta-analysis was conducted when more than two comparable studies were available for a specific outcome. For continuous data (i.e., pain scores), mean values and standard deviations, as reported in the included studies, were used to calculate the weighted mean difference (WMD) and the corresponding 95% confidence intervals (CIs). A random-effects model was applied a priori due to expected clinical and methodological heterogeneity among trials, using the restricted maximum likelihood (REML) estimator. Forest plots were generated to display the results. Cochran’s Q statistic and the I^2^ index were used to estimate heterogeneity across studies [34,35], with values of 25%, 50% and 75% representing, respectively, low, moderate and high heterogeneity [36]. Publication bias was assessed using Egger’s test when *n* ≥ 10 studies were included and visually inspected through funnel plots [37] (command “meta funnelplot”).

## 3. Results

### 3.1. Study Selection

A total of 5639 articles (279 in Embase, 2163 in PubMed and 3197 in Scopus) were retrieved through electronic search. Manual search through bibliographies of relevant reviews and trials led to 20 additional records. After removal of duplicates, 4874 articles were screened and 114 full-texts were assessed for eligibility. Finally, 12 studies were included and qualitatively analyzed, 4 of which were considered eligible for the quantitative synthesis of data (meta-analysis). Inter-reviewer agreement resulted in a Cohen’s kappa coefficient (κ) of 0.91. The process of study selection and the search results are summarized in Figure 1.

### 3.2. Study Characteristics

#### 3.2.1. Study Design and Study Population

Characteristics of the included trials are summarized in Table 1. A total of 12 randomized clinical trials were included, comprising 484 participants who received a single implant each. The follow-up period, starting from the implant placement, varied across studies, ranging from 6 months [38,39,40,41,42,43] to 36 months [44]. One trial assessed postoperative outcomes at 10 months [45] and three studies reported data at 12 months of follow-up [17,46,47].

In several trials, IIP was performed in the maxilla from premolar to premolar [46,47,48], while in two studies in both the maxillary and mandibular premolar-to-premolar areas [17,42]. Three trials were limited to maxillary premolar sites [40,41,44], and one study focused on mandibular premolars [38]. Two studies investigated exclusively the esthetic zone, either confined to incisors [39] or extended to incisors and canines [43].

#### 3.2.2. Type of Intervention

Individual study outcomes are summarized in Table 2. The deltas (Δ) between baseline and the last follow-up measurements are reported as presented in the original articles, or otherwise calculated by the reviewers, whenever feasible.

Xenografts

Seven trials evaluated the use of xenografts [17,39,40,44,45,46,47].

**Table 1 materials-18-05427-t001:** Characteristics of the included studies.

Author (Year)	Country (Setting)	Participants (N) Gender, Smoking	Mean Age (Year) Mean ± SD	Follow-Up in Months	Reason(s) for Extraction	Socket Morphology	Intervention(s)	Implant Characteristics	Timing and Type of Provisionalization	Measurements Methods	Outcomes	Conclusions
Xenograft												
Cardaropoli et al. (2014) [17]	Italy (Turin, Private Practice)	52 15 M, 11 F (Test) 14 M, 12 F (Control) Smokers: included smokers < 10 cig/day (requested to stop smoking before and after surgery)	42 ± 14 years (Test) 44 ± 13 years (Control)	12	Crown/root fracture, endodontic treatment failure, advanced caries	Second premolar to second premolar, with intact buccal plate (maxillary and mandibular)	Flapless extraction, with elevation of the interdental papillae only when necessary Bone-to-implant gap filled with DBBM blended with collagen and covered with a porcine-derived collagen membrane, secured with single sutures.	Brand: Biomet 3i (Osseotite Tapered Certain); Type: NR; Macrodesign: conical/tapered; Surface: Osseotite; Connection: NR; Length: 11.5–15 mm; Diameter: 3.25–5 mm; Position: crestal “flush”	Titanium healing abutment positioned at the end of the surgery Provisional crown cemented at 3 months	Cast-based measurement at the time of implant insertion (T0) and 12 months later (T12) Examiner calibration: NR	Survival rate, ridge width, ridge height	Ridge preservation with DBBM blended with collagen reduced soft tissue alterations and improved esthetic stability compared with IIP alone
Bittner et al. (2020) [47]	USA (Columbia Universy)	32 9 M, 23 F Smokers: excluded	52.3 ± 4 years Age range: 26–86 years	12	Caries, fracture, poor endodontic prognosis	First premolar to first premolar, with intact buccal bone plate (maxillary)	DBBM blended with collagen	Brand: Zimmer Biomet (Certain); Type: NR; Macrodesign: NR; Surface: NR; Connection: NR; Length: 8.5–15 mm; Diameter: 3.25–5.0 mm; Position: platform at the buccal crest or 1 mm below	Implants with Insertion torque > 20 N/cm restored with an immediate full-contour screw-retained provisional occlusion (*n* = 22). Implants with primary stability ≤ 20 Ncm restored with customized healing abutment (*n* = 10).	Clinical Measurements CBCT Digital Cast Analysis Examiner calibration and blinding: N	Horizontal and vertical soft tissue dimensional changes, post-operative complications	DBBM blended with collagen with immediate implants showed no major differences, except slightly less horizontal changes and better distal papilla stability at 12 months
Mastrangelo et al. (2018) [44]	Italy (Multicenter, private practice)	108 (102 analysed) 31 M, 20 F (Test) 32 M, 19 F (Control) Smokers: included; *n*: 48 %24.3	44 ± 6.7 years Age range: 18–72 years	36	NR	Premolars (maxillary)	Small full-thickness labial flaps were repositioned to completely cover the implant DBBM plus pericardium membrane	Brand: Dentaurum (tioLogic); Type: NR; Macrodesign: NR; Surface: NR; Connection: NR; Length: 11–13 mm; Diameter: 3.7–4.2 mm; Position: implant head 1–2 mm below the most apical bone peak; slightly palatal.	Standard titanium abutment and removable prosthesis	X-ray Clinical measurement Examiner calibration and blinding: both calibration and blinding applied	Survival rate, peri-implant marginal bone loss, PES, PPD post-operative complications	The use of an anorganic bovine bone substitute with a resorbable collagen barrier around the single immediate postextractive implant does not influences marginal bone loss and probing depth but seems to improve the esthetic outcomes after a 3-year follow-up.
Jacobs et al. (2020) [45]	USA (University of North Carolina)	33 8 (42%) M, 11 (58%) F (Test) 6 (43%) M, 8 (57%) F Smokers: excluded	53 ± 20 (Test) 65 ± 14 (Control)	10	NR	First premolar to first premolar, with no adjacent edentulous spaces (maxillary)	DBBM blended with collagen	Brand: Dentsply Sirona (OsseoSpeed TX Profile); Type: NR; Macrodesign: sloped-platform; Surface: OsseoSpeed; Connection: NR; Length: 11–15 mm; Diameter: 4.5 mm; Position: implant–abutment interface aligned with the facial osseous crest; ~2 mm palatal to the mucosal zenith	Bonded pontic Natural crown An interim acrylic single-tooth removable partial denture 12 weeks after implant placement screw-retained provisionalization	PVS impression of the maxillary anterior sextant Quantitative photographic analysis using standardized digital photographs CBCT Examiner calibration and blinding: blinding applied	PES Vertical and horizontal hard tissue changes Vertical soft tissue changes	Mucosal and hard tissue changes after flapless immediate placement of sloped-platform implants did not differ significantly whether or not DBBM blended with collagen was added to the facial extraction gap
Girlanda et al. (2019) [39]	Brasil (Dental Clinic of Paulista University)	22 4 M, 18 F Smokers: excluded	Age range: 21 to 58 years	6	NR	Incisors, with intact buccal alveolar wall (maxillary)	DBBM blended with collagen with a flapless approach	Brand: Biomet 3i (Full Osseotite Tapered Certain); Type: NR; Macrodesign: tapered; Surface: Full Osseotite; Connection: internal hexagon; Length: NR; Diameter: 4.1 mm; Position: implant shoulder 3 mm apical to the buccogingival margin; palatal position	Temporary abutment for 3 months, followed by fixed prosthesis	CBCT and 3D Imaging Clinical measurement with standardize stent Examiner calibration and blinding: both calibration and blinding applied	soft tissue height. buccal GAP bone tissue, width	In the anterior maxilla, flapless immediate implant placement with DBBM blended with collagen and immediate provisionalization resulted in superior preservation of both hard and soft tissues compared to sites without biomaterial
Grassi et al. (2019) [40]	Italy (University of Bari)	45 Flap-graft 6 M, 9 F Flap-nograft 6 M, 8 F Noflap-nograft 6 M, 9 F Smokers: included smokers < 10 cig/day	47.3 ± 12.9 years	6	NR	Premolars, with more than 4 mm of apical bone and no more than a 3 mm loss in the buccal bone plate (maxillary)	15 Flap-graft, 14 Flap-nograft 15 Noflap-nograft, residual space between the implant surface and socket walls grafted with cortical bone of equine origin, fully enzyme deantigenized	Brand: Bone System (“2P” Implant, Italy); Type: NR; Macrodesign: cylindrical; Surface: NR; Connection: NR; Length: 10–13.5 mm; Diameter: 4.1 mm; Position: implant shoulder 1 mm apical to the palatal marginal bone level	Healing screw placed at 6 months. Impressions for final restoration taken after 3 weeks, full zirconia crowns delivered 1 week later	CBCT Examiner calibration and blinding: blinding applied, measurements assessed by a a sigle experience examiner	Horizontal and vertical hard tissue dimensional changes, surgical intra-operative and post-operative complications Pain and discomfort (VAS)	Noflap-nograft surgery for post-extraction implants when sufficient buccal bone is present allows for a reduced and well-accepted surgical intervention that provides satisfactory results in terms of ‘jump space’ filling and dimensional bone preservation
Fettouh et al. (2023) [46]	Egypt (Cairo University)	40 4 (20%) M, 16 (80%) F (Test) 9 (50%) M, 11 (60%) F (Control) Smokers: excluded	37.4 ± 7.8 (Test) 35.5 ± 9.9 (Control)	12	NR	From premolar to premolar, thick gingival phenotype, intact but thin labial plate of bone extending 7 mm (≤1 mm) and intact palatal bone extending at least 6 mm apically	Flapless approach with DBBM	Brand: Bone System (“2P” Implant, Italy); Type: NR; Macrodesign: cylindrical; Surface: NR; Connection: NR; Length: 10–13.5 mm; Diameter: 4.1 mm; Position: implant shoulder 1 mm apical to the palatal marginal bone level	Immediate anatomical customized healing abutment Removable prosthesis with no interference with the soft tissue and no loading over the customized healing abutment.	CBCT Radiographic measurements of horizontal and vertical labial bone dimensions Examiner calibration and blinding: blinding applied	FMPS (%), FMBS (%), Pain VAS, Radiographic measurements of horizontal and vertical labial bone dimensions Horizontal labio-palatal bone width	Flapless IIP with or without DBBM in the labial gap is effective in preserving the alveolar bone dimension
Alloplastic graft												
Daif et al. (2013) [38]	Egypt (Cairo University)	28 18 F, 10 M Smokers: NR	34 years Age range: 22–48 years	6	Trauma, extensive dental caries, endodontic failure	Premolars (mandibular)	Porous beta-TCP	Brand: Zimmer (TSV); Type: bone-level; Macrodesign: tapered screw-vent; Surface: NR; Connection: NR; Length:11.5 or 13 mm; Diameter: 3.7 or 4.1 mm; Position: platform equicrestal or 1.5 mm subcrestal to the labial crest; labial gap ≥ 1.5 mm	Cover screw for 3 months Gingival former for 2 weeks	CT scan and 3D imaging Examiner calibration and blinding: NR	Bone density	Poor-phase multiporous beta-TPC seems to enhance the bone density when inserted into the gaps around immediate dental implants
Naji et al. (2021) [41]	Egypt (University of Mansoura)	48 Flap with graft 5 M, 11 F Flap without graft 7 M, 9 F Flapless without graft 6 M, 10 F Smokers: excluded	41.5 Age range: 28 to 55 years	6	NR	Premolars with intact socket walls, a horizontal gap > 2 mm in size, and a buccal bone plate thickness ≥ mm (maxillary)	16 Flap without graft, 14 flap with graft, 15 flapless without graft Alloplastic nanocrystalline calcium sulphate bone graft covered with an absorbable collagen membrane	Brand: Neo Biotech; Type: NR; Macrodesign: two-piece tapered screw-type, threaded; Surface: SLA; Connection: NR; Length: 11.5 or 13 mm; Diameter: 3.7 mm; Position: implant shoulder 1 mm subcrestal to the buccal crest	Healing abutments	CBCT Examiner calibration and blinding: blinding applied	Pain (NRS) Horizontal hard tissue dimensions	Flapless without graft shows similar buccal bone preservation as flap with graft when the bone plate is intact and gap > 2 mm
Mixed graft												
El Ebiary et al. (2023) [43]	Egypt (Cairo University)	24 3 M (25%) 9 F (75%) (Test) 6 M (50%) 6 F (50%) (Control) Smokers: NR	20 to 50 years Mean (SD) 30.5 (9.6) Test 33.8 (12.5) Control	6	Non-restorable upper anterior teeth	Incisors and canines, with completely intact labial plate and interproximal bone levels (maxillary)	50% DBBM and 50% autogenous bone	Brand: Implant Direct (Legacy); Type: NR; Macrodesign: tapered con buttress threads progressivi e tre cutting grooves; Surface: NR; Connection: NR; Length: 13–16 mm; Diameter: 3.7–4.2 mm; Position: traiettoria palatale; jumping gap 2–3 mm	3D-printed screwed temporary restoration	Clinical measurement Examiner calibration and blinding: NR	PES	Grafting the jumping distance utilizing the Dual Zone Grafting technique helps achieve a better esthetic outcom
Connective Tissue Graft (CTG)												
Guglielmi et al. (2022) [42]	Italy (Milan, Dental Department of San Raffaele Hospital)	30 17 F, 13 M Smokers: included smokers < 10 cig/day (requested to stop smoking before and after surgery)	53.4 ± 12.2 years Age range: 34–74 years)	6	root fracture, caries, root resorption, or endodontic failure.	From second premolar to second premolar with intact buccal plate or presenting a maximum of 3 mm of buccal dehiscence; The distance between interdental bone crest and buccal bone crest ≤ 3 mm after tooth extraction.	CTG The buccal flap was coronally advanced to tightly adapt to the healing abutment.	Brand: Winsix (KE; BioSAF IN Srl); Type: NR; Macrodesign: NR; Surface: NR; Connection: NR; Length: 9–15 mm; Diameter: 3.8 or 4.5 mm (endo-osseous 4.0/4.7 mm); Position: implant shoulder 1 mm apical to the buccal bone crest	healing abutments No implant-supported temporary restorations were used for the first 6 months.	Clinical measurements CBCT Soft tissue measurements (3D scanner) Examiner calibration and blinding: both calibration and blinding applied	KT width thickness of buccal bone wall (BC thick) S-IC, internal horizontal buccal gap dimension S-OC, horizontal buccal crest dimension R-B, vertical distance between the implant shoulder to top of the buccal bone crest. horizontal buccal bone resorption (HBBR) vertical buccal bone resorption (VBBR) osseous ridge width (ORW) Soft tissue contour Soft tissue thickness VAS score.	The adjunct of a CTG at the time of IIP, without bone grafting, does not influence vertical bone resorption. The use of CTG seems to reduce the horizontal changes of the alveolar ridge that occur.
Socket Shield Technique (SST)												
Venkatraman et al. (2023) [49]	India (New Delhi, Centre for Dental Education and Research)	22 14 (64%) M, 8 (36%) F Smokers: excluded	28.4 ± 4.7 years Age range: 18 to 45 years	12	Trauma, endodontic failure and unrestorable teeth	Incisor, with thick gingival phenotype and 3 to 5 mm of available bone apical to the existing root	SST	Brand: Adin (Touareg S); Type: NR; Macrodesign: NR; Surface: NR; Connection: NR; Length: NR; Diameter: NR; Position: implant placed 2 mm apical to the labial cortical bone level; palatal trajectory with 1–2 mm horizontal distance to the labial shield (SST) or to the labial wall (CT)	Immediate provisional screw-retained crowns fabricated on temporary abutments within 48 h of implant placement	Superimposition of 3D scans Clinical measurement Examiner calibration and blinding: blinding applied	Soft tissues dimensional changes PES	Volumetric changes in buccal soft tissue after IIP in the maxillary incisor area are inevitable, however such changes are reduced when using the SST

Abbreviations: NR, not reported; SD, standard deviation; KT, keratinized tissue; KTW, keratinized tissue width; PES, Pink Esthetic Score; FMPS, full-mouth plaque score; FMBS, full-mouth bleeding score; PPD, probing pocket depth; VAS, visual analogue scale; NRS, numeric rating scale; CBCT, cone-beam computed tomography; CEJ, cemento-enamel junction; FGM, free gingival margin; PA, periapical radiograph; STC, soft tissue contour; STT, soft tissue thickness; HBBR, horizontal buccal bone resorption; VBBR, vertical buccal bone resorption; ORW, osseous ridge width; ORR, osseous ridge reduction; SST, socket shield technique; CTG, connective tissue graft; T0/T6/T10/T12/T36, timepoints (baseline and months of follow-up).

Cardaropoli et al. (2014) [17] evaluated ridge dimensional changes over 12 months of follow-up and reported significantly lower deltas in the test (grafted) group (ridge width: 0.69 ± 0.68 mm; ridge height: −0.58 ± 0.76 mm) compared to the control one (ridge width: 1.92 ± 1.02 mm; ridge height: −1.69 ± 1.72 mm). Girlanda et al. (2019) [39] demonstrated significantly greater hard tissue width, measured 1 mm apical to the crest and in a buccolingual direction, in the “IIP + deproteinized bovine bone derived with collagen” group (6.57 ± 0.45 mm vs. 6.07 ± 0.24 mm).

Significantly lower vertical bone loss (0.1 ± 1.4 mm vs. 1.2 ± 0.7 mm) and reduced horizontal collapse (0.5 ± 0.4 mm vs. 0.8 ± 0.4 mm) was documented by Fettouh et al. (2023) [46] in the DBBM group, compared to negative controls.

Similarly, Grassi et al. (2019) [40] reported significantly lower horizontal buccal bone changes (−0.4 ± 0.8 mm) in the “flap-graft” group, in contrast to the “flap-no graft” one (−1.1 ± 0.9 mm), but did not find significant inter-group difference regarding vertical hard tissue dimensions at 6 months of follow-up.

Regarding soft tissue dynamics, Bittner et al. (2020) [47] observed more favorable vertical soft tissue dimensional changes in the “DBBM-C” group (Δ at distal papilla −0.4 ± 1.2 mm vs. −1.4 ± 1.1 mm; *p* = 0.02). Similarly, Girlanda et al. (2019) [39] found less vertical soft tissue height reduction at mesiobuccal and distobuccal sites in the test group.

Conversely, Mastrangelo et al. (2018) [44] and Jacobs et al. (2020) [45] reported comparable vertical hard and soft tissue changes between groups.

Esthetic scores, assessed through validated indices, were only recorded in two studies. Jacobs et al. (2020) [45] reported comparable PESs between groups (8.2 vs. 8.3), while Mastrangelo et al. (2018) [44] found significantly higher values in the sites where an anorganic bovine bone and resorbable collagen barrier were used (9.70 vs. 8.14).

In the study by Grassi et al. (2019) [40], the “no flap-no graft” group participants consistently reported significantly lower self-reported postoperative pain (53.2 ± 12.1 mm) compared to the “flap-graft” (71.4 ± 11.2 mm) and “flap-no graft” ones (75.1 ± 9.8 mm). Conversely, Fettouh et al. (2023) [46] found no significant inter-group differences regarding pain scores.

Alloplastic grafts

Two trials investigated alloplastic grafts [38,41]. Daif et al. (2013) [38] observed significantly greater bone density in the β-TCP graft group (1490 ± 358 HU) compared to the control group (1245 ± 165 HU) at 6 months of follow-up.

Significantly lower horizontal bone dimensional changes were reported by Naji et al. (2021) [41] in the “flap with graft” group, compared to the “flap without graft” one (0.37 ± 0.09 mm vs. 0.91 ± 0.54 mm, respectively). Moreover, pain scores at 7 days were significantly higher in the “flap with graft” group (5.14 ± 0.69) compared to the “flap without graft” (3.71 ± 0.76) and “flapless without graft” groups (0.71 ± 0.49).

Mixed grafts

A single trial assessed a mixed graft combining DBBM and autogenous bone [43]. A significant improvement in PESs was recorded in grafted sites from post-operative evaluation to 6 months follow-up (11.58 ± 1.16 to 12.42 ± 1.44). in contrast, non-grafted sites experienced a significant decrease in PES values (11.75 ± 1.71 to 11.17 ± 1.53). 

Connective tissue grafts (CTG)

One study examined the use of a CTG in combination with the immediate implant placement [42]. In grafted sites, more favorable horizontal ridge changes (−1.16 ± 0.5 mm vs. −2.09 ± 0.53 mm at 1 mm; *p* = 0.0003) and soft tissue contours deltas (−0.32 to −0.04 mm vs. −1.94 to −1.08 mm; *p* < 0.05) were recorded. Patient-reported postoperative morbidity scored significantly higher in the CTG group (2.73 ± 1.62) compared to the non-CTG group (1.07 ± 0.70).

**Table 2 materials-18-05427-t002:** Study outcomes.

Author (Year)	Implant Survival (%)	Hard Tissue Measurements (mm, Mean ± SD)	Ridge Dimensions (mm, Mean ± SD)	Soft Tissue Measurements	KT (mm, Mean ± SD)	PES (mm, Mean ± SD)	% FMPS, FMBS (mm, Mean ± SD)	PPD (mm, Mean ± SD)	Pain (VAS) (Mean ± SD)	Post op Complications (*n* of Cases)
**Xenograft**										
Cardaropoli et al. (2014) [17]	**Control Group**									
	96.15	NR	*Cast based measurements* *Ridge width:* ΔT0–12 months: 1.92 ± 1.02 * *Ridge height:* ΔT0–12 months: −1.69 ± 1.72 *	NR	NR	NR	NR	NR	NR	NR
	**Test Group**									
	100	NR	*Cast based measurements* *Ridge width:* ΔT0–12 months: 0.69 ± 0.68 * *Ridge height:* ΔT0–12 months: −0.58 ± 0.76 *	NR	NR	NR	NR	NR	NR	NR
Bittner et al. (2020) [47]	**Control Group**									
	100	NR	NR	*Casts digital superimposition* *(measurements at 3 and 4 mm from the T0-FGM)* Δ*horizontal dimensional changes T0–12 months:* 3 mm = −1.01 ± 0.45; 4 mm = −0.80 ± 0.33 *Stent measurement* Δ*vertical dimensional change T0–12 months:* *Mesial* −0.9 ± 1.4; *Distal* −1.4 ± 1.1 (*p* = 0.02) * *Buccal* −1.3 ± 1.5 Δ*soft tissue tickness (at 3, 4 and 8 mm from the T0–FGM) T0–12 months* 3 mm: 0.35 mm 4 mm: 0.40 mm 8 mm: −0.10 mm	NR	NR	NR	NR	NR	0
	**Test Group**									
	100	NR	NR	*Casts digital superimposition* *(measurements at 3 and 4 mm from the T0-FGM)* Δ*horizontal dimensional changes T0–12 months:* 3 mm = −0.84 ± 0.64; 4 mm = −0.64 ± 0.62 *Stent measurement* Δ*vertical dimensional change T0–12 months:* *Mesial* −0.6 ± 1.2 *Distal* −0.4 ± 1.2 * *Buccal* −0.9 ± 1.2 Δ*soft tissue tickness (at 3, 4 and 8 mm from the T0-FGM) T0*–*12 months* 3 mm: 0.70 mm 4 mm: 0.80 mm 8 mm: 0.50 mm	NR	NR	NR	NR	NR	0
Mastrangelo et al. (2018) [44]	**Control Group**									
	98.2	*PA xrays* Δ*marginal bone level* (T0–T36): –0.28 ± 0.304 *Mesial* ΔT0–T36: −0.32 ± 1.13 * *Distal* ΔT0–T36: −0.17 ± 1.09 *	NR	NR	NR	T36: 9.70 ± 2.02 *	ΔT0–T36: 1.40 ± 1.619	NR	NR	Poor surgical complications in both gropus (17 patients in total) At 36 months: inflammation in 56 patients, 2 peri-implantitis (no distinctions between groups)
	**Test Group**									
	98.3	*PA xrays* *Delta marginal bone level* (T0–T36): −0.25 ± 0.362 *Mesial* ΔT0–T36: −0.16 ± 0.66 * *Distal* ΔT0–T36: −0.24 ± 0.73 *	NR	NR	NR	T36: 8.14 ± 1.90 *	ΔT0–T36: 1.69 ± 1.345	NR	NR	
Jacobs et al. (2020) [45]	**Control Group**									
	97% (overall)	*CBCT measurements* *Vertical* *CEJ/Crown margin to crest* T0: 2.95 10 months: 2.26 ± 0.77 *Horizontal* *1 mm subcrestal* T0: 0.96 10 months: 1.47 ± 0.85 *Midimplant* T0: 0.84 10 months: 1.30 ± 1.28 *1 mm from apex* T0: 1.13 10 months: 1.90 ± 2.07	NR	*Clinical measurement* Δ*vertical dimensional change, T0*–*T10 months* *Mesial papilla* 0.57 ± 0.59 *Distal papilla* 0.79 ± 0.75 *Midfacial aspect* 0.92 ± 0.67	NR	10 months: 8.2 ± 1.8	NR	NR	NR	NR
	**Test Group**									
	97% (overall)	*CBCT measurements* *Vertical* *CEJ/Crown margin to crest* T0: 3.46 10 months: 2.02 ± 1.04 *Horizontal* *1 mm subcrestal* T0: 1.02 10 months: 1.63 ± 0.71 *Midimplant* T0: 0.99 10 months: 1.83 ± 1.17 *1 mm from apex* T0: 0.77 10 months: 2.54 ± 2.01	NR	*Clinical measurement* Δ*vertical dimensional change* *Mesial papilla* 0.33 ± 0.46 *Distal papilla* 0.49 ± 0.62 *Midfacial aspect* 0.94 ± 1.13	NR	10 months: 8.3 ± 2.5	NR	NR	NR	NR
Girlanda et al. (2019) [39]	Control Group									
	NR	*CBCT measurements* *Buccolingual measurement* *1 mm apical to crest* Baseline 6.75 ± 0.27; 6 months 6.07 ± 0.24 * *3 mm apical to crest* Baseline 6.83 ± 0.28; 6 months 6.15 ± 0.24 * *5 mm apical to crest* Baseline 6.86 ± 0.27; 6 months 6.18 ± 0.24 * *Buccal GAP* Baseline: 2.45 ± 0.52; 6 months 2.17 ± 0.46	NR	Clinical measuremen, *distance from the reference point on the stent to the gingival margin* *Vertical dimensional change* *Mesial* Baseline 8.55 ± 1.57; 6 months 9.55 ± 1.57 * *Distal* Baseline 8.36 ± 1.57; 6 months 9.35 ± 1.43 * *Buccal* Baseline 10.82 ± 1.54; 6 months 11.82 ± 1.54	NR	NR	NR	NR	NR	NR
	Test Group									
	NR	*CBCT measurements* *Buccolingual measurement* 1 mm apical to crest Baseline 7.04 ± 0.49; 6 months 6.57 ± 0.45 * *3 mm apical to crest* Baseline 7.12 ± 0.49; 6 months 6.65 ± 0.45 * *5 mm apical to crest* Baseline 7.15 ± 0.49; 6 months 6.69 ± 0.45 * *Buccal GAP* Baseline 2.55 ± 0.52; 6 months 2.29 ± 0.48	NR	Clinical measuremen, stent *Vertical dimensional change* *Mesial* Baseline 8.45 ± 1.81; 6 months 8.36 ± 1.75 * *Distal* Baseline 8.45 ± 1.75; 6 months 8.35 ± 1.54 * *Buccal* Baseline 11.82 ± 2.32; 6 months 11.82 ± 2.32	NR	NR	NR	NR	NR	NR
Grassi et al. (2019) [40]	**Control Group**									
	100	*CBCT measurements* *Horizontal bone level changes* A-EA Flap-nograft: ΔT0–T6 = −0.1 (1.2) Noflap-nograft: ΔT0–T6 = −0.3 (1.5) M-EM Flap-nograft: ΔT0–T6 = −1.2 (0.8) Noflap-nograft: ΔT0–T6 = −0.8 (0.8) B-EB Flap-nograft: post-surgery ΔT0–T6 = −1.1 (0.9) * Noflap-nograft: post-surgery ΔT0–T6 = −1.0 (1.1) B-IB Flap-nograft: post-surgery ΔT0–T6 = −1.8 (0.6) Noflap-nograft: post-surgery ΔT0–T6 = −2.0 (0.9) *Vertical bone level changes* *C-P* Flap-nograft: post-surgery ΔT0–T6 = −0.2 (0.6) Noflap-nograft: post-surgery ΔT0–T6= −0.1 (0.6)	NR	NR	NR	NR	NR	NR	Flap-nograft 4 h = 45.3 ± 10.3 * 24 h = 75.1 ± 9.8 * 3 days = 29.8 ± 10.8 * 7 days = 9.3 ± 4.9 * Noflap-nograft 4 h = 35.2 ± 13.4 * 24 h = 53.2 ± 12.1 * 3 days = 18.2 ± 11.3 * 7 days = 5.4 ± 3.8 *	NR
	**Test Group**									
	100	*CBCT measurements* *Horizontal bone level changes* A-EA Flap-graft: post-surgery ΔT0–T6 = −0.2 (1.1) M-EM Flap-graft: post-surgery ΔT0–T6 = −0.4 (0.7) B-EB Flap-graft: post-surgery ΔT0–T6 = −0.4 (0.8)* B-IB Flap-graft: post-surgery ΔT0–T6 = −2.3 (0.8) *Vertical bone level changes* C-P Flap-graft: post-surgery ΔT0–T6 = −0.3 (0.7)	NR	NR	NR	NR	NR	NR	Flap-graft 4 h = 49.5 ± 15.2 * 24 h = 71.4 ± 11.2 * 3 days = 31.6 ± 9.8 * 7 days = 10.2 ± 5.3 *	NR
Fettouh et al. (2023) [46]	**Control Group**									
	100	*CBCT measurements* *Horizontal labial bone thickness* (at three levels below the labial bone crest) ΔT0–1 year: 0 mm: 2.1 ± 1.5 2 mm: 1.9 ± 1.5 5 mm: 1.6 ± 1.6 Δ*Labio-palatal bone width T0-1 year* 0 mm: −0.7 ± 2.05 2 mm: −0.7 ± 2.05 5 mm: −0.8 ± 2.19 *Vertical Crestal bone level change,* Δ*T0*–*1year*: −1.2 ± 0.7	NR	NR	NR	NR	FMPS (%) Baseline: 17.1 ± 2.7 1 year: 11.8 ± 3.4 FMBS (%) Baseline: 9.6 ± 2.1 1 year: 6.6 ± 2.6	NR	VAS 24 h: 61.8 ± 9.5 3 days: 41.8 ± 6.3 7 days: 5.8 ± 3.5	NR
	**Test Group**									
	100	CBCT measurements *Horizontal* *labial bone plate thickness (at three levels below the labial bone crest)* Delta Baseline—1 year: 0 mm: 1.7 ± 1.10 2 mm: 1.7 ± 1.20 5 mm: 1.7 ± 1.30 Labio-palatal bone width Δ*T0*–*1year* (0 mm):−0.6 ± 0.5 (2 mm): −0.6 ± 0.3 (5 mm): −0.5 ± 0.4 *Vertical Crestal bone level change,* Δ*T0*–*1year:* −0.1 ± 1.4	NR	NR	NR	NR	FMPS (%) Baseline: 16.3 ± 1.9 1 year: 11.9 ± 1.9 FMBS (%) Baseline: 9.8 ± 2.0 1 year: 7.4 ± 2.0	NR	VAS 24 h: 60 ± 10.9 3 days: 39.3 ± 7.8 7 days: 5.8 ± 3.5	NR
**Alloplastic graft**										
Daif et al. (2013) [38]	**Control Group**									
	100	*Bone density (Hounsfield units) CT scan* 6 months: 1245 ± 165 *	NR	NR	NR	NR	NR	NR	NR	0
	**Test Group**									
	100	*Bone density (Hounsfield units) CT scan* 6 months: 1490 ± 358 *	NR	NR	NR	NR	NR	NR	NR	2 (Mild soft tissue infection)
Naji et al. (2021) [41]	**Control Group**									
	100	*CBCT measurements* *Horizontal dimension of buccal alveolar bone* Flap without graft ΔT0–T6 = 0.91 ± 0.54 * Flapless without graft ΔT0–T6 = 0.24 ± 0.11 *	NR	NR	NR	NR	NR	NR	7 days (NRS) Flap without graft: 3.71 ± 0.76 * Flapless without graft: 0.71 ± 0.49 *	NR
	**Test Group**									
	100	*CBCT measurements* *Horizontal dimension of buccal alveolar bone* Flap with graft ΔT0–T6 = 0.37 ± 0.09 *	NR	NR	NR	NR	NR	NR	7 days (NRS) Flap with graft: 5.14 ± 0.69 *	NR
**Mixed graft**										
El Ebiary et al. (2023) [43]	**Control Group**									
	NR	NR	NR	NR	NR	Immediate post-operative: 11.75 ± 1.71 6 months: 11.17 ± 1.53 *	NR	NR	NR	NR
	**Test Group**									
	NR	NR	NR	NR	NR	Immediate post-operative: 11.58 ± 1.16 6 months: 12.42 ± 1.44 *	NR	NR	NR	NR
**Connective Tissue Graft** **(CTG)**										
Guglielmi et al. (2022) [42]	**Control Group**									
	100	*CBCT measurements* *Horizontal Bone Dimensions, HBBR 1–5 mm below the most coronal point of the buccal osseous ridge)* HBBR 1: −1.59 ± 0.54 mm * HBBR 2: −1.13 ± 0.40 mm * HBBR 3: −0.96 ± 0.37 mm * HBBR 4: −0.79 ± 0.34 mm * HBBR 5: −0.78 ± 0.34 mm * *Vertical Bone Dimensions* VBBR: −0.66 (0.75) mm	*CBCT measurements* *Osseous ridge width (1–5 mm below the most coronal point of the buccal osseous ridge)* ORR 1 (mm): −2.09 ± 0.53 * ORR 2 (mm): −1.66 ± 0.79 ORR 3 (mm): −1.45 ± 0.76 ORR 4 (mm): −1.13 ± 0.59 ORR 5 (mm): −1.06 ± 0.49	*3D scanner measurements* *Buccal soft tissue contour changes (*Δ*STC baseline-6 months) (at 1, 2, 3, 4, and 5 mm apical to the baseline gingival margin)* STC 1: −1.94 ± 0.83 * STC 2: −1.85 ± 0.78 * STC 3: −1.57 ± 0.63 * STC 4: −1.30 ± 0.64 * STC 5: −1.08 ± 0.80 * STC Volume (mm^3^;): 0.16 ± 0.42 * *Soft tissue thickness (*Δ*STT baseline-6 months) (at 1, 2, 3, 4, and 5 mm respectively apical to the baseline and 6 M gingival margin)* STT 1: −0.16 ± 0.61 * STT 2: 0.12 ± 0.70 * STT 3: 0.81 ± 0.96 * STT 4: 0.88 ± 0.88 * STT 5: 0.11 ± 0.76 *	KTW, 6 months Mean KTW: 3.64 ± 1.29 mm Gain in KTW: 0.14 ± 1.28 mm	NR	NR	NR	VAS, 1 week 1.07 ± 0.70 *	0
	**Test Group**									
	100	*CBCT measurements* *Horizontal Bone Dimensions, HBBR 1–5 mm below the most coronal point of the buccal osseous ridge)* HBBR 1: −1.36 ± 1.17 mm HBBR 2: −0.89 ± 0.70 mm HBBR 3: −0.73 ± 0.53 mm HBBR 4: −0.69 ± 0.39 mm HBBR 5: −0.66 ± 0.45 mm *Vertical Bone Dimensions* VBBR: −0.66 (0.53) mm	*CBCT measurements* *Osseous ridge width (1–5 mm below the most coronal point of the buccal osseous ridge)* ORR 1 (mm): −1.16 ± 0.50 * ORR 2 (mm): −1.03 ± 0.66 ORR 3 (mm): −0.95 ± 0.60 ORR 4 (mm): −1.05 ± 0.68 ORR 5 (mm): −0.94 ± 0.56	*3D scanner measurements* *Buccal soft tissue contour changes (*Δ*STC baseline-6 months) (at 1, 2, 3, 4, and 5 mm apical to the baseline gingival margin)* STC 1: −0.32 ± 0.97 * STC 2: −0.04 ± 0.74 * STC 3: 0.11 ± 0.66 * STC 4: 0.18 ± 0.70 * STC 5: 0.13 ± 0.81 * STC Volume (mm^3^): 6.76 ± 8.94 * *Soft tissue thickness (*Δ*STT baseline-6 months) (at 1, 2, 3, 4, and 5 mm respectively apical to the baseline and 6 M gingival margin)* STT 1: 1.47 ± 1.08 * STT 2: 2.04 ± 1.18 * STT 3: 2.42 ± 1.63 * STT 4: 2.07 ± 1.22 * STT 5: 1.33 ± 1.17 *	KTW, 6 months Mean KTW: 4.53 ± 1.36 mm Gain in KTW: 0.60 ± 1.71 mm	NR	NR	NR	VAS, 1 week: 2.73 ± 1.62 *	0
**Socket Shield Technique** **(SST)**										
Venkatraman et al. (2023) [49]	**Control Group**									
	100	NR	NR	*3D scanner measurements, soft tissue volumentric change* T0–T12 months: −0.643 ± 0.35 mm *	NR	T0: 8.91 ± 1.30 T12: 11.18 ± 1.40 *	NR	NR	NR	0
	**Test Group**									
	100	NR	NR	*3D scanner measurements, soft tissue volumentric change* T0–T12 months: −0.1520 ± 0.86 mm *	NR	T0: 9.73 ± 1.27 T12: 12.91 ± 0.83 *	NR	NR	NR	0

Abbreviations: NR, not reported; SD, standard deviation; KT, keratinized tissue; KTW, keratinized tissue width; PES, Pink Esthetic Score; FMPS, full-mouth plaque score; FMBS, full-mouth bleeding score; PPD, probing pocket depth; VAS, visual analogue scale; NRS, numeric rating scale; CBCT, cone-beam computed tomography; CEJ, cemento-enamel junction; FGM, free gingival margin; PA, periapical radiograph; STC, soft tissue contour; STT, soft tissue thickness; HBBR, horizontal buccal bone resorption; VBBR, vertical buccal bone resorption; ORR, osseous ridge reduction; T0/T6/T10/T12/T36, timepoints (baseline and months of follow-up); Δ, change from baseline. Unit notes: FMPS/FMBS are percentages, not mm; PES is a score (not mm). Light grey shading is used to distinguish the test group from the control group. “*” indicates that, regarding that specific measurement, the difference between the test group and the control group is statistically significant.

Socket shield technique (SST)

SST was evaluated in one trial [49], which reported significantly lower soft tissue dimensional changes in the test group (−0.15 ± 0.86 mm), compared to the control one (−0.64 ± 0.35 mm) at 1 year of follow-up. A similar trend was observed for PESs (12.9 ± 0.8 vs. 11.2 ± 1.4).

Whenever reported, implant survival rates were consistently high, ranging between 96% and 100% at the reported follow-up periods, with no differences attributable to the adjunctive approach used. Surgical complications were rarely reported and, when described, were minor.

### 3.3. Quantitative Synthesis

For comparability across studies, pain scores originally reported on a 0–10 mm VAS were converted to a 0–100 mm scale [41,42]. In addition, the numerical rating scale (NRS) reported by Naji et al. (2021) [41] was considered comparable to a VAS and included in the quantitative synthesis after conversion. In studies where pain scores were recorded daily [40,46] the peak value was included in the meta-analysis to allow comparison with studies that reported pain only at a single postoperative time point, typically at 1 week [41,42].

Four studies [40,41,42,46] assessing patient-reported post-operative pain were quantitatively analyzed (Figure 2). The pooled data estimate demonstrated a statistically significant mean difference in favor of the control group (19.45 mm, 95% CI 0.55 to 38.36; *p* = 0.04).

In accordance with the threshold described by Higgins et al. (2003) [36], high heterogeneity was detected across studies (I^2^ = 96.9%, *p* < 0.001).

### 3.4. Risk of Bias Quality Assessment

The risk of bias assessment of individual studies is displayed in Figure 3. Overall, only two studies were rated at low risk of bias, reflecting a moderate to high degree of methodological limitations identified. Four studies were judged as having a high overall risk of bias due to negative ratings assigned to Domain 4 (limitations in the measurement of the outcome), Domain 2 (deviations from the intended interventions) and, in one case, due to missing outcome data (Domain 3).

Intention-to-treat analyses were rarely conducted; consequently, 10 out of the 12 included studies were rated as having a moderate to high risk of bias in Domain 2 (deviations from the intended interventions). On the other hand, only two studies were judged at moderate risk of bias for selective reporting of outcomes (domain 5).

### 3.5. Risk of Bias Across Studies

Given the *n* < 10 studies included in the quantitative synthesis, publication bias was only graphically ascertained, and the corresponding Funnel plot is presented in Appendix A. The scattered distribution should be interpreted cautiously due to the limited number of studies.

### 3.6. Certainty of Evidence

Among the 12 RCTs included, the implant survival rate consistently ranged between 96% and 100% in both test and control groups, leading to a moderate level of certainty, downgraded for indirectness due to short-term follow-up and small sample sizes. The esthetic outcomes assessed clinically (PES/WES) showed a mean improvement of approximately +1.0 to +1.7 points in grafted or augmented groups compared with controls, although the evidence was graded as low certainty because of heterogeneity in study design, outcome definitions, and non-standardized esthetic indices. Patient-reported esthetic outcomes (PROMs) were not consistently evaluated and were therefore rated as very low certainty.

For hard and soft tissue dimensional outcomes, the use of adjunctive materials such as xenografts or connective tissue grafts resulted in moderate reductions in peri-implant bone and soft tissue remodeling (mean horizontal ridge loss reduced by ≈1 mm compared with controls), yet the overall certainty was low due to heterogeneity of interventions and small study samples. Keratinized tissue width (KTW) increased modestly in CTG-treated sites (mean gain ≈ 0.6 mm), supported by moderate-certainty evidence. In contrast, postoperative morbidity, mainly assessed through visual analog scales (VASs), was consistently higher in test groups, with a pooled mean difference of +19.45 mm (95% CI 0.55–38.36; *p* = 0.04), resulting in moderate-certainty evidence downgraded for inconsistency (I^2^ ≈ 97%).

Overall, the body of evidence provides low-to-moderate certainty that adjunctive procedures may improve soft and hard tissue stability and esthetic outcomes following immediate implant placement, without affecting implant survival, but at the cost of increased short-term morbidity (Table 3).

## 4. Discussion

The present review systematically evaluates the available evidence on the clinical performance of biomaterials, grafting materials, and techniques in IIP, in terms of implant survival, professionally determined and patient-reported esthetic outcomes, peri-implant hard and soft tissues dynamics, and self-reported short-term post-operative pain. Twelve RCTs were included, four of which contributed to the quantitative synthesis. The findings suggest that immediate implants, while technically challenging, represent a safe and highly effective therapeutic approach. Esthetic outcomes and soft and hard tissue stability showed improvements when adjunctive techniques were applied. Within the biological framework of IIP healing, biomaterials appear to mitigate hard tissue remodeling, which in turn supports improved postoperative soft tissue stability. The observation that various types of supplementary materials yield comparable outcomes suggests that the critical factor lies in their ability to stabilize the blood clot within the gap through osteoconductive mechanisms. However, adjunctive procedures were associated with higher short-term post-operative morbidity. The results of the present systematic review should be interpreted with caution, since several included trials were judged at moderate or high risk of bias, mainly due to deviations from intended interventions and limitations in outcome assessment.

Implant survival rates ranged from 96.1% to 100% across the included studies, irrespective of the adjunctive procedure employed. This finding is in line with previous evidence showing that, when specific clinical conditions are met, immediate implant placement achieves survival rates between 97.3% and 99% after 2 years [1], although more recent data have reported a slightly higher risk of early failures compared to placement in healed sites [50]. In addition, postoperative complications, although recorded in a few trials [38,41,42,43,44], were negligible or entirely absent.

Six trials investigated deproteinized bovine bone mineral (DBBM) with or without collagen membranes [17,39,40,44,45,46,47], while one utilized cortical bone of equine origin. Results were heterogeneous: some trials demonstrated reduced ridge resorption and improved peri-implant tissue stability when DBBM was combined with collagen membranes or provisionalization [17,46], whereas others found no significant effect on marginal bone loss or soft tissue dimensions, despite improvements in esthetic outcomes with higher PES values [44,45,47]. Grassi et al. (2019) suggested that flapless non-grafted sites with sufficient buccal bone perform similarly to grafted sites [35,37,38,40]. Similarly, a recent systematic review by Zaki et al. (2021) documented reduced ridge resorption and improved esthetics when IIP was associated with the use of xenografts [24]. The widespread adoption of xenografts as adjunctive materials in immediate implant placement protocols reported in the literature is attributable to their clinical convenience in terms of hard and soft tissue changes [24,51,52]. Evidence on other biomaterials is more limited. Alloplastic substitutes have a positive impact on hard tissue outcomes [38,41], and a mixed graft combining DBBM and autogenous bone showed higher PES values [43].

One included trial investigated soft tissue augmentation: Guglielmi et al. (2022) showed that adding CTG during immediate implant placement did not significantly influence vertical bone resorption but reduced horizontal alveolar ridge contraction and improved soft tissue contour preservation [42]. Despite higher postoperative morbidity compared to controls, CTG was overall well tolerated and improved tissue stability in the esthetic zone. Seyssens et al. (2021) showed that CTG in IIP significantly improved midfacial soft tissue stability, reducing vertical changes by 0.41 mm and lowering the risk of ≥1 mm recession twelve-fold, though without benefits in marginal bone levels or PES [22], findings consistent with Guglielmi et al. (2022) [42]. More recently, Lambert et al. (2025) confirmed that soft tissue augmentation procedures reduce midfacial recession but failed to highlight advantages in clinician-assessed esthetic outcomes or patient-reported measures [26]. Other systematic reviews report similar results, showing gains in mucosal thickness and recession control, but no significant improvements in overall esthetic scores or patient satisfaction [25].

The quantitative synthesis of self-reported post-operative morbidity indicated that adjunctive procedures were associated with significantly higher pain scores compared to controls (WMD 19.45; 95% CI 0.55–38.36; *p* = 0.04). Three of the included trials consistently reported greater pain in the test groups [40,41,42], whereas one study did not find inter-group differences [46]. It should be noted that the substantial heterogeneity observed (I^2^ = 96.9%, *p* < 0.001) reflects the methodological variability across the included studies, in the type of grafting material used (xenograft, alloplasts and CTG) and in the surgical protocols applied. Grassi et al. (2019) [40], Naji et al. (2021) [41], and Fettouh adopted a flapless approach, while Guglielmi et al. (2022) [42] recorded postoperative pain following a surgical procedure involving a flap approach and a connective tissue graft harvesting. Although not associated with higher pain scores, these differences, which intuitively influence surgical patient experience, contributed to the variability between studies. Moreover, given the inclusion of only four randomized controlled trials this meta-analysis should be considered exploratory. Within the limitations of the quantitative synthesis, these findings suggest that adjunctive interventions, which may add complexity to the IIP protocol, are associated with higher short-term patient-reported discomfort. However, it should be noted that the reported pain scores are short-lived and unlikely to compromise long-term patient satisfaction or clinical outcomes. While adjunctive procedures may transiently increase postoperative morbidity, their potential benefits in terms of tissue stability and esthetic outcomes may compensate for this short-term drawback, especially in esthetically demanding sites.

A quantitative synthesis of hard and soft tissue changes as well as esthetic outcomes was not feasible due to the lack of standardization and comparability in the reported measurements. Methodological heterogeneity was also observed among the included trials, encompassing variations in grafting materials, surgical approaches (flapless versus flap), implant macro design and positioning, socket morphology, and soft tissue phenotype.

Moreover, the present review could not address a direct comparison between immediate implant placement with and without provisionalization, in the absence of other adjunctive gap-filling materials, focusing on soft tissue outcomes. Such comparison would have been clinically meaningful, since immediate provisionalization has been suggested to promote midfacial soft tissue stability [53,54], and prosthetic factors such as crown contour have also been shown to influence peri-implant soft tissue levels [55]. Future studies should specifically evaluate the role of provisionalization on peri-implant soft tissue behavior in the short and long term.

A distinctive strength of the present review lies in its strict methodological approach, in which IIP with adjunctive procedures was systematically compared to immediate implant placement alone, since, beyond general clinical recommendations for the use of biomaterials in IIP, there is no recognized gold standard. This choice has brought to light a critical aspect that is often overlooked in the “guilty pleasure” of adding regenerative strategies to a protocol that already demonstrates very high survival rates. The rationale behind adjunctive procedures is primarily to enhance esthetic results, however clinician-assessed esthetic outcomes are rarely assessed, and even more rarely in a standardized and reproducible manner. For instance, the PES [56,57] was evaluated in only four of the included trials [43,44,45,49]. Neither the papilla index nor the white esthetic score [57] was reported, and soft tissue assessments such as midfacial/buccal recession have not been consistently measured across studies. These objective parameters should be linked to subjective outcomes (PROMs), as they are inherently interrelated, to determine whether the clinical potential of adjunctive strategies in IIP outweighs its drawbacks, in terms of patient discomfort and morbidity. However, patient-reported outcomes, such as satisfaction with esthetic results and the overall treatment experience, are insufficiently and inconsistently reported. As a result, the clinical benefits of biomaterials and grafting techniques are being judged without adequately addressing the patient-centered dimension, which should be regarded as a fundamental endpoint when evaluating immediate implant placement in the esthetic zone.

### Implications for Future Research

The findings of the present systematic review pinpoint the need for standardized outcome reporting in trials investigating the clinical benefits of the use of adjunctive techniques in IIP.

Future studies should employ standardized and reproducible methods for assessing hard and soft tissue dimensional changes following IIP with different adjunctive approaches, to allow comparisons. The use of different imaging modalities, measurement tools and non-standardized clinical assessments currently hampers comparability and meaningful meta-analyses.

Clinician-assessed esthetic outcomes should be systematically evaluated, utilizing validated indices such as the Pink Esthetic Score (PES) and the White Esthetic Score (WES).

Patient-reported esthetic scores, overall treatment experience, postoperative morbidity and complications, including the frequency and type of medications taken during the postoperative period, should become an integral part of outcome assessments in clinical trials on IIP.

Moreover, adopting postoperative follow-ups and assessments should follow uniform timings, to facilitate accurate comparisons across trials.

## 5. Conclusions

Within the limitations of the present systematic review and meta-analysis, the findings support the evidence that IIP is associated with high survival rates. Although the adjunctive use of biomaterials, grafting materials and techniques may offer advantages in terms of peri-implant hard and soft tissue outcomes, it appears to be associated with greater short-term post-operative discomfort. Future research should incorporate standardized reporting of clinical, clinician-assessed and patient-reported outcome measures to clarify whether the additional morbidity and costs associated with adjunctive approaches are outweighed by their esthetic and functional benefits.

## Figures and Tables

**Figure 1 materials-18-05427-f001:**
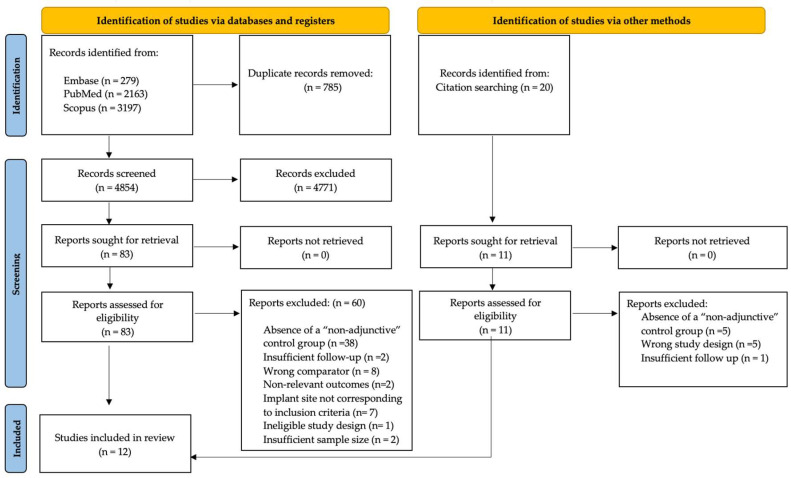
PRISMA 2020 flow diagram. Source: [29].

**Figure 2 materials-18-05427-f002:**
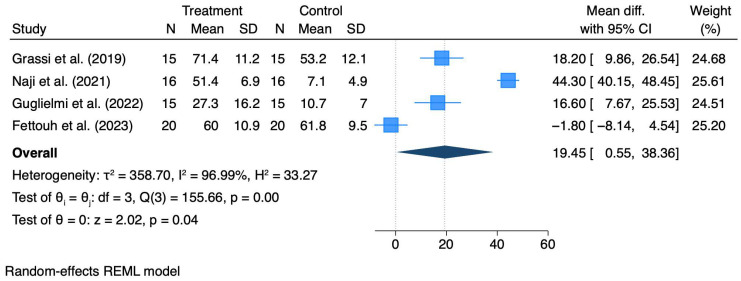
Forrest plot for Pain Scores. Studies included: Grassi et al. [40], Naji et al. [41], Guglielmi et al. [42], Fettouh et al. [46].

**Figure 3 materials-18-05427-f003:**
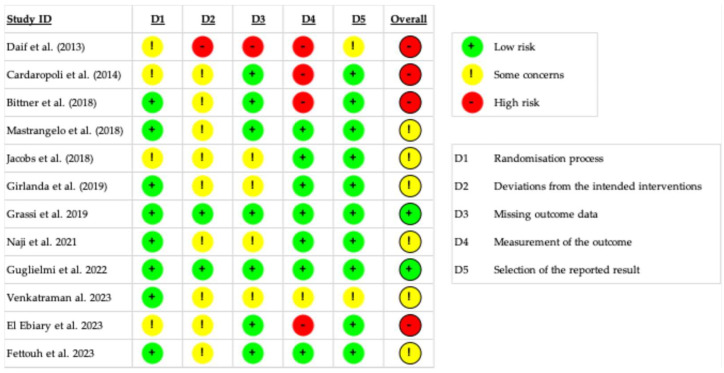
Risk of bias assessment of the included studies. Studies included: Daif et al. [38], Cardaropoli et al. [17], Bittner et al. [47], Mastrangelo et al. [44], Jacobs et al. [45], Girlanda et al. [39], Grassi et al. [40], Naji et al. [41], Guglielmi et al. [42], Venkatraman et al. [49], El Ebiary et al. [43], Fettouh et al. [46].

**Table 3 materials-18-05427-t003:** Level of certainty/quality of the evidence from RCTs assessed by Grading of Recommendations, Assessment, Development, and Evaluations (GRADE).

Outcome	№ of Participants (Studies)	Effect (95% CI or Mean ± SD)	Quality of Evidence (GRADE)	Summary/Comments
**Implant survival rate**	484 (12 RCTs)	96–100% in both groups; no significant differences	Moderate ⨁⨁⨁◯	Stable survival across all biomaterials; downgraded for indirectness (short follow-up, small samples).
**Clinically assessed esthetic outcomes (PES/WES)**	164 (4 RCTs)	PES improved by +1.0–1.7 points (Mastrangelo 2018 + 1.56; El Ebiary 2023 + 1.25; SST + 1.7)	Low ⨁⨁◯◯	Significant in some studies but inconsistent; non-standardized scoring; esthetic benefit possible but uncertain.
**Patient-reported esthetic outcomes (PROMs)**	— (none with validated PROMs)	Not reported	Very Low ⨁◯◯◯	No validated PROMs; serious indirectness and reporting bias.
**Hard tissue dimensional changes**	340 (8 RCTs)	Xenografts: Δridge width −0.7 mm vs. −1.9 mm (control); CTG: ΔHBBR −1.36 mm vs. −1.59 mm	Low ⨁⨁◯◯	Moderate ridge preservation with grafts; downgraded for heterogeneity and small samples.
**Soft tissue dimensional changes/contour stability**	212 (5 RCTs)	CTG: Δcontour −0.3 mm vs. −1.9 mm (control); SST: Δvolume −0.15 mm vs. −0.64 mm (control)	Low ⨁⨁◯◯	Consistent trend toward contour improvement; limited sample sizes and variable methodology.
**Keratinized tissue width (KTW)**	60 (2 RCTs)	CTG: KTW 4.53 ± 1.36 mm vs. 3.64 ± 1.29 mm (control); gain ≈ +0.6 mm	Moderate ⨁⨁⨁◯	KTW gain limited to CTG studies; reliable CBCT/3D measurements but small sample.
**Post-operative morbidity (pain, VAS/NRS)**	150 (4 RCTs, meta-analysis)	MD = +19.45 mm VAS (95% CI 0.55–38.36; *p* = 0.04); e.g., Grassi 2019 24 h: 71.4 ± 11.2 vs. 53.2 ± 12.1 mm	Moderate ⨁⨁⨁◯	Adjunctive procedures increase short-term pain; heterogeneity high (I^2^ ≈ 97%) but direction consistent.

Abbreviations: PES = Pink Esthetic Score; WES = White Esthetic Score; PROMs = Patient-Reported Outcome Measures; CTG = Connective Tissue Graft; SST = Socket Shield Technique; KTW = Keratinized Tissue Width; VAS = Visual Analogue Scale; CBCT = Cone-Beam Computed Tomography; RCT = Randomized Controlled Trial.

## Data Availability

No new data were created or analyzed in this study. Data sharing is not applicable to this article.

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
