# Peer review of "Adjunctive Procedures in Immediate Implant Placement: Necessity or Option? A Systematic Review and Meta-Analysis"

_materials, 2025, doi:10.3390/ma18235427_

Round 1

Reviewer 1 Report

Comments and Suggestions for Authors

In this effort, the authors have conducted a meta-analysis of adjunctive procedures in immediate implant placement. While the introduction, selected portions of the methods, and the discussion are generally adequate, the following revisions are recommended to enhance clarity, rigor, and overall presentation quality:

The abstract and portions of the methods appear to have been directly copied from an external source (possibly a registry or website). These sections should be rewritten in a cohesive narrative style, ensuring that all information is fully integrated into the manuscript text. This will improve readability and demonstrate a clear understanding of the methodology.

Table 2 is missing from the manuscript, and there is no accompanying degree of freedom (df) analysis for the funnel plot. Without this, readers cannot assess whether sufficient data exist to justify the funnel plot. Furthermore, based on the traffic light plot, only two analyses are identified as low-risk. The authors should clarify which analyses were included in the funnel plot and provide a rationale for their selection.

The forest plot also requires substantial improvement in formatting and labeling. In addition, the discussion of heterogeneity is currently superficial and should be expanded to address sources of variability and their potential impact on the pooled results.

To strengthen the study and statistical power, the authors are encouraged to broaden the scope to include a wider class of implants or adjunctive techniques. This would increase the degrees of freedom available for statistical analysis and enhance the robustness of the meta-analytic conclusions.

Author Response

Authors’ response to Reviewers’ comments

Journal:               Materials (ISSN 1996-1944)

Manuscript ID:  materials-3974649

Title of Paper:   Adjunctive procedures in immediate implant placement: necessity or option? A systematic review and meta-analysis         

Authors:              Isabella De Rubertis, Adriano Fratini, Maria Clotilde Carra, Marco Annunziata, Nicola Discepoli

Section:              Biomaterials

Special issue:     Biomaterials in Periodontology and Implant Dentistry

Date Sent:           15th November , 2025

We sincerely thank the Reviewers for their constructive evaluation of our manuscript, which guided us in improving the overall quality of the work. We hope that this revised version meets the standards required for publication in Materials.

Reviewer 1

“In this effort, the authors have conducted a meta-analysis of adjunctive procedures in immediate implant placement. While the introduction, selected portions of the methods, and the discussion are generally adequate, the following revisions are recommended to enhance clarity, rigor, and overall presentation quality”:

COMMENT 1

The abstract and portions of the methods appear to have been directly copied from an external source (possibly a registry or website). These sections should be rewritten in a cohesive narrative style, ensuring that all information is fully integrated into the manuscript text. This will improve readability and demonstrate a clear understanding of the methodology.”

Response: Thank you for your suggestion. According to your comment, the indicated sections have been revised to improve readability and to ensure consistency of style throughout the manuscript.

COMMENT 2

“Table 2 is missing from the manuscript, and there is no accompanying degree of freedom (df) analysis for the funnel plot. Without this, readers cannot assess whether sufficient data exist to justify the funnel plot. Furthermore, based on the traffic light plot, only two analyses are identified as low-risk. The authors should clarify which analyses were included in the funnel plot and provide a rationale for their selection.

The forest plot also requires substantial improvement in formatting and labeling.”

Response: Thank you for your comment and precise suggestion.

  1. Tables (Table 1, Table 2 and Table S1) were submitted as separate documents.
  2. All the studies included in the meta-analysis (n=4) were used to generate the funnel plot. We do recognize that the interpretation of Egger’s test results was not reliable with only 4 studies, therefore, we removed it from the revised manuscript. However, we retained the Funnel plot for descriptive purposes only, highlighting that it should be interpreted with caution.
  3. The Forest plot was produced using the latest version of the meta command in Stata IC 18. We have revised it, adding the degree of freedom and the vertical reference lines (line of no effect and overall effect line) to enhance clarity and readability.

COMMENT 3

“In addition, the discussion of heterogeneity is currently superficial and should be expanded to address sources of variability and their potential impact on the pooled results.”

Response: Thank you for your indication. According to your comment, the Discussion section of the revised manuscript has been implemented as follows: It should be noted that the substantial heterogeneity observed (I² = 96.9%, p < 0.001) reflects the methodological variability across the included studies, in the type of grafting material used (xenograft, alloplasts and CTG) and in the surgical protocols applied. Grassi et al. (2019), Naji et al. (2021), and Fettouh adopted a flapless approach, while Guglielmi et al (2023) recorded postoperative pain following a surgical procedure involving a flap approach and a connective tissue graft harvesting. Although not associated with higher pain scores, these differences, which intuitively influence surgical patient experience, contributed to the variability between studies. Moreover, given the inclusion of only four randomized controlled trials this meta-analysis should be considered exploratory.”

COMMENT 4

“To strengthen the study and statistical power, the authors are encouraged to broaden the scope to include a wider class of implants or adjunctive techniques. This would increase the degrees of freedom available for statistical analysis and enhance the robustness of the meta-analytic conclusions.”

Response: We thank you for your suggestion. We acknowledge that the inclusion criteria established a priori for the present systematic review were strict. However, we would like to emphasize that no limits were applied to the adjunctive techniques considered eligible for inclusion. Conversely, we maintained consistency regarding the eligible comparators, as we deliberately excluded the studies in which the control arm involved any intervention other than immediate implant placement alone. This approach greatly reduced the number of included studies. However, our objective was to assess the current state of art with the highest possible methodological rigor and internal validity, accepting a lower degree of external validity and applicability.

Reviewer 2 Report

Comments and Suggestions for Authors

Overall, this manuscript presents a methodologically well-structured and clinically relevant meta-analysis that addresses an important gap in implant research. With minor improvements in the clarity of discussion and visualization of key concepts, the study will offer stronger educational and translational value to both clinical and research audiences.

- The methodological design of this review broadly follows PRISMA 2020 and Cochrane guidance; however, several key details require clarification to ensure analytical transparency and reproducibility. The PROSPERO registration is appropriate, though the authors should specify whether any protocol amendments occurred and confirm adherence to the registered plan. The PICO structure is sound, yet the description of interventions lacks sufficient detail regarding graft material types and categories, which may limit subgroup interpretability. Measurement calibration or inter-operator reliability should be discussed to address potential bias in outcome assessment.

- The eligibility criteria need stronger justification: the English-only restriction introduces potential language bias, and the ≥10 participants per arm cutoff appears arbitrary. Similarly, the minimum 6-month follow-up may not capture long-term implant survival outcomes and should be reconsidered or discussed as a limitation. The literature search strategy would benefit from greater transparency, including a complete search string for at least one database and explicit search period coverage. Reporting of Cohen’s kappa values for inter-reviewer agreement is also expected.

- For statistical synthesis, the random-effects approach is appropriate but the estimation method (e.g., DerSimonian–Laird or REML) should be stated. Clarify when WMD versus SMD was applied, how median data were converted, and how binary outcomes were handled. Reporting heterogeneity with I² confidence intervals and specifying thresholds is recommended. The use of Egger’s test should be justified or limited to analyses with ≥10 studies. Finally, sensitivity/subgroup analyses and an assessment of evidence certainty (e.g., GRADE) are missing and should be included for completeness.

- The discussion effectively highlights the need for standardized and reproducible outcome reporting in immediate implant placement research; however, this central concept could be more clearly emphasized. I recommend that the authors visually or structurally reinforce the implications of a “standardized, reproducible protocol” — for example, by adding a concise schematic figure or an independent subsection summarizing its anticipated clinical and research advantages (e.g., enhanced comparability, reduced methodological bias, and improved patient-centered translation). Such an addition would improve reader comprehension and underscore the review’s translational relevance. Currently, the argument remains textually embedded within the discussion and conclusion; elevating it as a distinct visual or conceptual element would strengthen the manuscript’s educational and practical value.

Author Response

Authors’ response to Reviewers’ comments

Journal:               Materials (ISSN 1996-1944)

Manuscript ID:  materials-3974649

Title of Paper:   Adjunctive procedures in immediate implant placement: necessity or option? A systematic review and meta-analysis         

Authors:              Isabella De Rubertis, Adriano Fratini, Maria Clotilde Carra, Marco Annunziata, Nicola Discepoli

Section:              Biomaterials

Special issue:     Biomaterials in Periodontology and Implant Dentistry

Date Sent:           15th November , 2025

We sincerely thank the Reviewers for their constructive evaluation of our manuscript, which guided us in improving the overall quality of the work. We hope that this revised version meets the standards required for publication in Materials.

Reviewer 2

“Overall, this manuscript presents a methodologically well-structured and clinically relevant meta-analysis that addresses an important gap in implant research. With minor improvements in the clarity of discussion and visualization of key concepts, the study will offer stronger educational and translational value to both clinical and research audiences.”

COMMENT 1

“The methodological design of this review broadly follows PRISMA 2020 and Cochrane guidance; however, several key details require clarification to ensure analytical transparency and reproducibility. The PROSPERO registration is appropriate, though the authors should specify whether any protocol amendments occurred and confirm adherence to the registered plan. The PICO structure is sound, yet the description of interventions lacks sufficient detail regarding graft material types and categories, which may limit subgroup interpretability. Measurement calibration or inter-operator reliability should be discussed to address potential bias in outcome assessment.”

Response: Thank you for your helpful suggestions. The Materials and Methods section has been implemented to include the missing details according to your comment and the PRISMA 2020 checklist has been updated accordingly.

We confirm that no major amendments have been made to the originally submitted PROSPERO protocol.

Regarding the “Intervention” component of the PICO framework, we recognize that it may appear broad, however our intention was to impose no limits to the adjunctive techniques used in IIP, and to compare them with cases where no adjunctive approaches were used at all. According to the retrieved results, following the established inclusion criteria, we grouped them, whenever possible, according to the category of adjunctive approach.

Finally, we agree with you about the fact that it might be useful to include information about examiners’ calibration and this information has been included in Table 1.

COMMENT 2

“The eligibility criteria need stronger justification: the English-only restriction introduces potential language bias, and the ≥10 participants per arm cutoff appears arbitrary. Similarly, the minimum 6-month follow-up may not capture long-term implant survival outcomes and should be reconsidered or discussed as a limitation. The literature search strategy would benefit from greater transparency, including a complete search string for at least one database and explicit search period coverage. Reporting of Cohen’s kappa values for inter-reviewer agreement is also expected.”

Response: Thank you for your comment. We appreciate the chance to further explain the choice behind the established inclusion criteria. The ≥10-participant inclusion criterion was adopted in order to ensure adequate study power and to exclude small pilot or split-mouth trials with insufficient sample sizes. The choice to only include English-language publications was made to reduce errors in data interpretation and in light of the findings of a recent systematic review, demonstrating that English-only restrictions have a negligible impact on results and conclusions of medical topics reviews (Dobrescu et al., 2021).

Moreover, the minimum 6-month follow-up, without a maximum follow up restrcition, was in line with the aim of assessing how long the effect of an adjunctive approach may last. It is well established that approximately two-thirds of the biological changes occurring in post-extraction sockets- affecting esthetic as well as hard and soft tissue outcomes- take place within the first 3-6 months, continuing up to the 12 months (Schropp et al., 2007).

We have now included in Appendix 1 the full search strings utilized for the electronic database search.

Regarding the indication of the inter-examiner agreement measurement, we clarified in the revised manuscript that the k score reported in paragraph 3.1 refers to the Cohen’s kappa coefficient (κ).

COMMENT 3

“For statistical synthesis, the random-effects approach is appropriate, but the estimation method (e.g., DerSimonian–Laird or REML) should be stated. Clarify when WMD versus SMD was applied, how median data were converted, and how binary outcomes were handled. Reporting heterogeneity with I² confidence intervals and specifying thresholds is recommended. The use of Egger’s test should be justified or limited to analyses with ≥10 studies. Finally, sensitivity/subgroup analyses and an assessment of evidence certainty (e.g., GRADE) are missing and should be included for completeness.”

Response: Thank you for your comment. The estimation method used was restricted maximum likelihood (REML) (command meta summarize, random(reml)). The pooled effect was expressed as a weighted mean difference (WMD). Median values were not included, and no binary outcomes were analyzed. We agree about the fact that the results of Egger’s test are not reliable, given the number of included studies in the present review. We therefore removed the description of Egger’s test results from the results section. The funnel plot was included for descriptive and explotatory purposes, specifying it should be interpreted with caution.

Finally, in accordance to your comment, the summary of the level of certainty/quality of the evidence from randomized controlled trials assessed by GRADE was incorporated in the revised manuscript.

COMMENT 4

“The discussion effectively highlights the need for standardized and reproducible outcome reporting in immediate implant placement research; however, this central concept could be more clearly emphasized. I recommend that the authors visually or structurally reinforce the implications of a “standardized, reproducible protocol” — for example, by adding a concise schematic figure or an independent subsection summarizing its anticipated clinical and research advantages (e.g., enhanced comparability, reduced methodological bias, and improved patient-centered translation).

Such an addition would improve reader comprehension and underscore the review’s translational relevance. Currently, the argument remains textually embedded within the discussion and conclusion; elevating it as a distinct visual or conceptual element would strengthen the manuscript’s educational and practical value.”

Response: Thank you for your suggestion. We have addedd a paragrafh at the end of the Discussion section summarizing the implications for future reseach in the light of the obtained results. The paragraph reads as follows: “The findings of the present systematic review pinpoint the need for standardized outcome reporting in trials investigating the clinical benefits of the use of adjunctive techniques in IIP.

Future studies should employ standardized and reproducible methods for assessing hard and soft tissue dimensional changes following IIP with different adjunctive approaches, to allow comparisons. The use of different imaging modalities, measurement tools and non-standardized clinical assessments currently hampers comparability and meaningful meta-analyses.

Clinician-assessed esthetic outcomes should be systematically evaluated, utilizing validated indices such as the Pink Esthetic Score (PES) and the White Esthetic Score (WES).

Patient-reported esthetic scores, overall treatment experience, postoperative morbidity and complications, including the frequency and type of medications taken during the postoperative period, should become an integral part of outcome assessments in clinical trials on IIP.

Moreover, adopting postoperative follow-ups and assessments should follow uniform timings, to facilitate accurate comparisons across trials.”

Although we did not add a new schematic figure at this stage, this subsection is concise and clearly structured in bullet-like form within the narrative, as suggested, making the key concept more visible and educational for readers.

References

Dobrescu, A. I., Nussbaumer-Streit, B., Klerings, I., Wagner, G., Persad, E., Sommer, I., Herkner, H., & Gartlehner, G. (2021). Restricting evidence syntheses of interventions to English-language publications is a viable methodological shortcut for most medical topics: a systematic review. Journal of clinical epidemiology137, 209–217. https://doi.org/10.1016/j.jclinepi.2021.04.012

Schropp, L., Wenzel, A., Kostopoulos, L., & Karring, T. (2003). Bone healing and soft tissue contour changes following single-tooth extraction: a clinical and radiographic 12-month prospective study. The International journal of periodontics & restorative dentistry23(4), 313–323.

Round 2

Reviewer 1 Report

Comments and Suggestions for Authors

Dear Authors,

Thank you for addressing the comments.

Furthermore, gaining access to the tables made a crucial difference in understanding the original scientific intent.

Author Response

Authors’ response to Reviewers’ comments

Journal:               Materials (ISSN 1996-1944)

Manuscript ID:  materials-3974649

Title of Paper:   Adjunctive procedures in immediate implant placement: necessity or option? A systematic review and meta-analysis         

Authors:              Isabella De Rubertis, Adriano Fratini, Maria Clotilde Carra, Marco Annunziata, Nicola Discepoli

Section:              Biomaterials

Special issue:     Biomaterials in Periodontology and Implant Dentistry

Date Sent:           19/11/2025

We express our gratitude for the efforts of the Referees, whose contributions enhanced the quality of this manuscript. We have responded to the minor revisions received, and we hope that this version aligns with the specifications of Materials.

Reviewer 1

“Dear Authors,

Thank you for addressing the comments.

Furthermore, gaining access to the tables made a crucial difference in understanding the original scientific intent.”

Response: We thank you for the feedback and constructive suggestions provided to improve the current version of the manuscript. We addressed the minor revisions marked as “Can be improved”.  

Reviewer 2 Report

Comments and Suggestions for Authors

Although the authors have added substantial material in this revision round, they appear to have responded to my previous comments in a purely mechanical manner. The newly added references and methodological descriptions are not accompanied by explanations of "why" each element was incorporated. The authors should demonstrate greater ownership of their work and more carefully evaluate which suggestions to adopt or rebut, rather than simply implementing all comments as instructed. In this process, it would be appropriate to present a clearer and more decisive schematic for the key claims, developed with deeper consideration.

Author Response

Authors’ response to Reviewers’ comments

Journal:               Materials (ISSN 1996-1944)

Manuscript ID:  materials-3974649

Title of Paper:   Adjunctive procedures in immediate implant placement: necessity or option? A systematic review and meta-analysis         

Authors:              Isabella De Rubertis, Adriano Fratini, Maria Clotilde Carra, Marco Annunziata, Nicola Discepoli

Section:              Biomaterials

Special issue:     Biomaterials in Periodontology and Implant Dentistry

Date Sent:           19/11/2025

We express our gratitude for the efforts of the Referees, whose contributions enhanced the quality of this manuscript. We have responded to the minor revisions received, and we hope that this version aligns with the specifications of Materials.

Reviewer 2

“Although the authors have added substantial material in this revision round, they appear to have responded to my previous comments in a purely mechanical manner. The newly added references and methodological descriptions are not accompanied by explanations of "why" each element was incorporated. The authors should demonstrate greater ownership of their work and more carefully evaluate which suggestions to adopt or rebut, rather than simply implementing all comments as instructed. In this process, it would be appropriate to present a clearer and more decisive schematic for the key claims, developed with deeper consideration.”

Response: We thank you for the additional evaluation and for the opportunity to further refine the quality of our work. We would also like to clarify that the amendments included, following the previous round of revisions, resulted from a deliberate and thoughtful assessment of the Reviewers’ suggestions. We adopted only those recommendations that were consistent with the scope and objectives of our study and that we considered scientifically justified and methodologically sound. We respectfully emphasize that this selection process was intentional and grounded in scientific judgment, rather than in uncritical compliance.

We acknowledge the importance of clearly articulating the rationale behind our methodological and conceptual choices, as well as, more importantly, asserting their ownership. Our intention in the previous round of revisions was to ensure that every suggestion, in line with our perspective, was properly addressed. More in detail, regarding the GRADE assessment, we had not initially considered it, as we did not believe it was necessary to assess the overall quality of the evidence; we considered that a between- and across-study assessment of bias, in addition to the initial filter provided by the selection of RCTs, was sufficient. However, we decided to reconsider our initial evaluation and, as justified also in the main text, we implemented the review according to your appreciated suggestion.

With regard to the other inputs we received, as can be seen from the response letter, we did not accept everything that was proposed and, to support our position in these cases, we justified our choices with appropriate references.

Additionally, we revised the sections marked as “Can be improved” and we trust that this version more clearly reflects the scientific rationale and ownership of our work, and we hope it satisfactorily addresses the Reviewer’s concerns.